# Welfare Implications of Low-Dose Atipamezole Reversal of Tiletamine/Zolazepam/Xylazine Anaesthesia in Pigs

**DOI:** 10.3390/ani15020258

**Published:** 2025-01-17

**Authors:** Rachel Layton, David S. Beggs, Andrew Fisher, Peter Mansell, Daniel Layton, Peter A. Durr, Teegan Allen, Grace Taylor, Michael L. Kelly, David T. Williams, Kelly J. Stanger

**Affiliations:** 1CSIRO, Australian Centre for Disease Preparedness, Geelong, VIC 3219, Australia; peter.durr@csiro.au (P.A.D.); teegan.allen@csiro.au (T.A.); grace.taylor@csiro.au (G.T.); mike.kelly@csiro.au (M.L.K.); d.williams@csiro.au (D.T.W.); kelly.stanger@csiro.au (K.J.S.); 2Melbourne Veterinary School, Faculty of Science, University of Melbourne, Werribee, VIC 3030, Australia; dbeggs@unimelb.edu.au (D.S.B.); pmansell@unimelb.edu.au (P.M.); 3CSIRO, Health and Biosecurity, Geelong, VIC 3219, Australia; daniel.layton@csiro.au

**Keywords:** anaesthesia, zoletil, behaviour, research, swine

## Abstract

Pigs are routinely used in many areas of research. Anaesthesia is commonly used in research to protect pig welfare, safely collect research samples, and achieve study objectives. Atipamezole is an anaesthetic reversal agent that is used to partially reverse the effects of anaesthesia, leading to shorter recovery times. However, anaesthetic reversal agents have also been shown to cause side effects and turbulent anaesthetic recoveries in pigs. The present study compared the recovery of anaesthetised pigs both with and without a low dose of atipamezole for the partial reversal of anaesthesia. We aimed to determine if using atipamezole reversal at a dose below the published range reduced recovery times and the physiological impacts of anaesthesia without introducing side effects and adverse behaviours. We found that while low-dose atipamezole decreased anaesthetic recovery time and improved the maintenance of core temperature in most anaesthetised pigs, it introduced and exacerbated a range of side effects and behaviours that could lead to poorer welfare outcomes for research pigs. This study provides new information on the welfare impacts of using a low dose of atipamezole for the reversal of pig anaesthesia, which will allow for the refinement of anaesthesia regimes in laboratory pigs.

## 1. Introduction

Pig models are increasingly used in scientific research, as their similar anatomical structure to humans and importance in food production makes them valuable models for translational and agricultural studies [1]. Achieving scientific objectives in these studies is commonly contingent on the collection of blood and other biological samples (such as swabs), which requires effective pig restraint for successful sample collection and staff safety [2]. Effective restraint of laboratory pigs for sample collection is even more crucial in the case of zoonotic disease studies, where infected pigs and the use of sharps presents a high risk to staff [3].

Restraint of laboratory pigs relies on manual (e.g., snaring), mechanical (e.g., pig sling with positive reinforcement training, vascular access ports) or chemical methods (anaesthesia or sedation) [4]. Snares provide a means of manual restraint to enable minimally invasive procedures and sample collection without the use of anaesthesia [5]. However, the stress associated with snaring can lead to aversion behaviour, making repeated restraint for sampling very challenging in a laboratory environment [6]. Additionally, stress induced by snaring causes disruption to physiological and immunological responses, which can lead to reduced animal welfare and increased variability in study data [7,8,9]. An alternative restraint method for the repeated sampling of laboratory pigs is the implementation of a positive reinforcement training regime and sling restraint, which facilitates voluntary sample collection [10]. Pigs can be trained to use a sling restraint to allow for the collection of blood directly from the vein [11]. However, this method presents an additional risk of sharps injury to operators and is likely to be an unacceptable risk for some studies, such as those conducted under high containment conditions [12]. In scenarios where repeated blood samples are required, the surgical implantation of vascular access ports (VAPs) can facilitate safe, voluntary blood collection from trained pigs [13]. However, the use of VAPs is only warranted when the study duration or number of blood collections is sufficient to justify the invasive surgical procedure and where access to sufficiently trained surgical staff is available to facilitate VAP implantation. Where VAPs are not appropriate, anaesthetising pigs can provide a safe alternative means of blood collection [12].

Anaesthesia can be useful for a range of procedures in pigs, including blood collection, restraint for imaging studies, and wound management [1,10]. Whilst anaesthetising laboratory pigs for sample collection and procedures is arguably warranted in many instances, anaesthesia can cause a range of physiological, immunological, and cognitive disruptions that can have adverse effects on animal welfare and physiology [9]. Among these physiological disruptions are hypertension [14,15], hypotension [16], decreased heart rate [15,16], decreased respiratory rate [16], and reduced Sp02% [14,15,16]. The use of anaesthesia may also alter experimental outcomes, as found in a study by Nash (2021) [17]. This research demonstrated that mice anaesthetised with ketamine/xylazine were more susceptible to a low dose of influenza virus than mice that were not anaesthetised [17]. Furthermore, Guo et al. 2023 conducted a scoping review to determine the effects of general anaesthesia on cognitive function in rodent animal models. The authors found that cognitive deficits were reported up to 14 days post-anaesthesia in 77% of studies (*n* = 82/106), occurring more frequently in studies of older animals after prolonged inhalational anaesthetics [18]. Despite this, a lack of anaesthetic use to alleviate pain and distress can also have repercussions on experimental outcomes, cause safety concerns, and negatively impact animal welfare [9,19]. Consequently, while the appropriate use of anaesthesia can be a crucial component of research with animals, minimising the duration and impacts of anaesthesia to prevent adverse welfare and physiological consequences is also important.

General anaesthetic combinations for pigs in research typically include combinations of tiletamine–zolazepam or ketamine with xylazine or medetomidine due to its small volume, reliable immobilisation, and short half-life and duration [16,20,21]. The administration of α2-adrenergic receptor antagonists (e.g., yohimbine and atipamezole) can antagonise the effects of α2-adrenergic agonists, reducing recovery times via receptor competition [22]. Yohimbine and atipamezole can be used for the partial antagonism of α2-adrenergic agonist combination anaesthesia in pigs. Both antagonist agents are known to be effective at decreasing the time to recovery and physiological impacts of anaesthesia but can cause adverse side effects and excitable, turbulent recoveries [23,24]. This can more commonly occur when an anaesthetic combination contains an α2-adrenergic agonist and a dissociative anaesthetic (e.g., tiletamine, ketamine) for the provision of adequate restraint and anaesthetic depth. The antagonism of the α2-adrenergic agonist does not affect the dissociative anaesthetics, which remain in circulation and may increase the incidence of turbulent recoveries [25].

The present study was conducted as part of a larger study of pigs vaccinated and challenged with Japanese encephalitis virus (JEV), undergoing regular xylazine and tiletamine/zolazepam (XTZ) anaesthesia for sample collection. The recovery of pigs anaesthetised with XTZ was assessed with and without the use of low-dose atipamezole. We hypothesised that using a lower dose of atipamezole than is standard in the published literature would reduce recovery time and the physiological effects of anaesthesia, without introducing additional adverse side effects and behaviours of potential risk to animal welfare.

## 2. Materials and Methods

### 2.1. Animal Management and Acclimation

Eight female 6-week-old Landrace cross pigs were sourced from a commercial piggery located near Geelong, Victoria, and housed four per room within the microbiologically secure animal facility at ACDP. All pigs were weighed (21–25 kg) and health checked on arrival prior to allocation to the Japanese encephalitis virus (JEV) vaccine study. The health check on arrival consisted of measuring respiratory rate and heart rate, assessment of skin, ocular and oral health, presence or absence of scouring or abnormal faecal output, and body condition score. Each group of four pigs were maintained in a pen measuring 3 m × 2.5 m. Each pen was furnished with toys that were changed daily, a plastic bed containing straw, a rubber mat, and a feed trough containing Barastoc™ pig grower pellets (Ridley Corporation, Melbourne, Australia) that pigs could access ad libitum. A small amount of fruit and vegetable was provided daily for variety and enrichment. Room temperature was maintained at 22 °C, and lights were maintained on an 8 h light/16 h dark cycle. Upon arrival at the facility, pigs were housed in these conditions for 14 days to acclimate prior to the JEV study.

During anaesthetic recoveries, pigs were moved onto a dry rubber mat once anaesthetic depth was reached (lateral recumbency with the absence of spontaneous movement, approximately 5 min after anaesthetic administration) to aid heat retention and provide a cushioned surface. Throughout the study, all pigs remained healthy, did not display any clinical signs of disease, and did not shed virus via oral swab, nasal swab, or blood collected daily for eight days after viral administration

### 2.2. Experimental Design

All pigs were anaesthetised on two occasions, one week apart, within a broader JEV vaccine study. No atipamezole was administered during recovery from the first (control) anaesthetic event (CTL), and a low dose of atipamezole was administered during recovery from the second anaesthetic event (ATZ). All pigs underwent the same vaccination and challenge regime. A study timeline relevant to the assessment of anaesthesia recovery is presented in Table 1.

### 2.3. Anaesthetic and Antagonist Regime

Zoletil™ (tiletamine 50 mg/mL and zolazepam 50 mg/mL) (Virbac Animal Health, Carros, France) at 4.4 mg/kg and xylazine 2.2 mg/kg was administered intramuscularly (XTZ). Pigs were aged 16 weeks and weighed in the range of 48–55 kg at the first assessment period and aged 17 weeks and weighed 50–57 kg at the second assessment period. Individual weights were rounded up to the nearest kg for the determination of anaesthetic and antagonist dose calculations. After the completion of sampling procedures during the ATZ session, atipamezole at 0.12 mg/kg was administered intramuscularly 15 min after anaesthetic injection. This was equivalent to a ratio of 1:18 atipamezole to xylazine. All intramuscular injections were delivered using a 16 g 1 ½” needle into the pelvic limb caudal to the stifle into the biceps femoris. Pigs were placed onto a shared rubber mat in a divided 1.5 m × 2.5 m section of their home pen for sample collection and recovery. As pigs became ambulatory, they were encouraged outside of the recovery area so as not to injure or interfere with the recovery of other pigs.

### 2.4. Assessment of Anaesthesia Recovery

Pigs were assessed at 5 min intervals, commencing immediately prior to anaesthetic administration. A single operator conducted all clinical assessments and recovery scores at 5 min intervals (Table 2), whilst a second operator conducted all behavioural assessments) for 60 s at 5 min intervals (Table 3). Each 60 s period was video recorded and retrospectively reviewed to ensure that real-time behaviour assessments were accurate. A recovery score between 0 and 5 was assigned to pigs at each assessment point, based on the recovery score system developed for the assessment of pig anaesthesia recovery by Nishimura et al. (1992) and described in Table 2 [23]. Recovery from anaesthesia was defined as two consecutive recovery scores of 0, which marked the end of the assessment period.

### 2.5. Determination of Consciousness

Pigs were deemed to have regained consciousness at the point of the first instance of a return of executive functioning (e.g., return of functions necessary for control of behaviours) and the likely point of the return of mental cognition (e.g., mental processes including thinking, perception, and memory). In the absence of specialised equipment (e.g., electroencephalography), assessment of the return of executive function and cognition was determined by behavioural observations of response to stimuli and purposeful movements (equivalent to a recovery score of 3).

### 2.6. Assessment of Clinical Variables

Assessments of selected clinical variables were conducted to determine the physiological state of pigs throughout recovery. Respiratory rate (visual assessment of breaths per minute), respiratory effort (visual assessment of normal, slightly increased, or notably increased), palpebral reflex by touching the medial canthus of the eye (fully present, partially present as a slower response and incomplete closure of the eyelid, or absent), pedal withdrawal response by pinch stimulus between the toes (present or absent), and rectal temperature were all measured every 5 min. Pulse rate (beats per minute palpated via the saphenous artery) was measured at each timepoint except immediately prior to anaesthesia and at the point of recovery where manual pulse rate assessment was not reliable or safe due to pig movement. Reference ranges for normal pig temperature, pulse rate, and respiratory rate were used as described by Sipos (2013) [26].

### 2.7. Behavioural Assessment

To assess the welfare of pigs during recovery from anaesthesia, all displayed behaviours were recorded within each 60 s assessment period every 5 min (Table 3).

Behaviours and recovery characteristics that occurred at least partially prior to the postulated return of consciousness and those that were not likely to result in injury or lasting harm were deemed unlikely to cause distress. These recovery characteristics and behaviours were, therefore, classified as having a low welfare impact and were recorded as either present or absent during the 60 s monitoring windows. Behaviours that were displayed solely after the postulated return to consciousness (recovery score of 3, 2, 1, or 0) and were likely to cause injury were deemed to cause distress and, therefore, have a negative effect on pig welfare. These behaviours were, therefore, classified as having a high welfare impact and were categorised as mild, moderate, or severe based on the number of times recorded in each 60 s assessment period, as described in Table 4.

Categorisation of behaviours with a high welfare impact into mild, moderate, and severe was conducted retrospectively, based on the range of frequency of each behaviour exhibited by pigs. Rare frequency of the behaviour was classified as mild presentation, occasional frequency was classified as moderate presentation, and persistent frequency was classified as severe presentation.

### 2.8. Statistical Analysis

Statistical analysis was performed using GraphPad Prism (version 9.1.2, La Jolla, CA, USA) for preliminary descriptive and exploratory analyses and R v. 4.4.2 for formal confirmatory analyses. Time to recovery and temperature data were confirmed to be normally distributed via both the D’Agostino and Pearson test and the Shapiro–Wilk test (alpha = 0.05). Confirmatory analyses were undertaken to determine whether the effect of atipamezole on the pigs’ post-anaesthesia Score 3 recovery times and the rectal temperature at these times were statistically significant. For both analyses, the experiments were determined to have a simple within-subjects design and a paired *t*-test was used for analysis as implemented with the R-function “t.test” from the “stats” library, with the level of statistical significance set to be 0.05. Differences between sessions for time to recovery (time to reach recovery score 3) and physiological metrics were analysed using multiple paired *t*-tests by comparing the two sessions at each 5 min timepoint for each metric measured. Behaviours and recovery characteristics were reported descriptively and did not undergo statistical analysis.

## 3. Results

### 3.1. Low-Dose Atipamezole Leads to More Rapid Recovery from Anaesthesia

To determine the impact of low-dose atipamezole on the recovery of pigs anaesthetised with xylazine/tiletamine/Zolazepam (XTZ), the recovery of pigs from anaesthesia with and without a low dose of atipamezole was assessed. The median time taken to achieve a recovery score of 3 (return to consciousness) was 50.0 min for the control (CTL) recovery and 32.5 min for the atipamezole (ATZ) recovery. This difference was statistically significant (*p* = 0.038) (Figure 1). However, not all pigs had a decreased recovery time during the ATZ recovery, with two out of eight pigs showing an increase (Figure 1). Once pigs had returned to consciousness, there were no further significant differences in progression to full standing recovery between CTL and ATZ anaesthesia events (*p* ≥ 0.05) (Appendix A). During the ATZ recovery, pigs also showed a faster return of palpebral and pedal withdrawal reflexes (*p* ≤ 0.01) (Appendix A) and an increased number of eye blinks (*p* ≤ 0.05) (Appendix A) prior to return to consciousness.

### 3.2. Low-Dose Atipamezole Improves Thermoregulation During Anaesthetic Recovery

The median rectal temperature when pigs returned to consciousness was 38.0 °C during the CTL recovery and 38.6 °C during the ATZ recovery, a difference which was highly significant (*p* = 0.006) (Figure 2). As was observed for time to recovery, the effect of low-dose atipamezole on rectal temperature was not uniform, with one out of eight pigs showing a slightly lower temperature at recovery compared to when atipamezole was not administered (Figure 2). During the CTL recovery, seven out of eight pigs had rectal temperatures below the normal lower limit of 38.5 °C at the point of returning to consciousness. In comparison, two out of eight pigs had temperatures below the normal limit at return to consciousness during the ATZ recovery (Figure 2). Differences in rectal temperature between the CTL and ATZ recoveries continued after the return to consciousness (Appendix A). No differences were observed between anaesthesia recovery events for pulse rate or respiratory effort (Appendix A), and respiratory rates remained mostly within normal limits for both anaesthesia recovery events (Appendix A).

### 3.3. Low-Dose Atipamezole Increases Adverse Behaviours During Anaesthesia Recovery

There were no low-impact negative behaviours or side effects noted during the CTL recovery, compared with 45 instances during the ATZ recovery (Figure 3). Negative behaviours included mouth chewing, excess salivation, leg tremors, and leg paddling, all of which were displayed by pigs during the ATZ recovery both prior to and after the return to consciousness (Figure 3).

Behaviours that occurred solely after the return to consciousness and could likely cause injury or distress were determined to have a high welfare impact (head rearing and dropping, falls from sitting or standing, bar biting, and forceful nosing). During the CTL recovery, pigs displayed seven mild instances of behaviours with a postulated high welfare impact and no moderate or severe instances (Figure 4). Conversely, during the ATZ recovery, pigs displayed 24 mild, 11 moderate, and 6 severe instances of behaviours with a postulated high welfare impact (Figure 4). During the CTL recovery, three out of eight pigs displayed at least one high-impact behaviour, compared to seven out of eight pigs during the ATZ recovery (Figure 4).

## 4. Discussion

This study is the first to assess the recovery of pigs anaesthetised with xylazine/tiletamine/zolazepam (XTZ), with and without the use of a low dose of atipamezole. An atipamezole to xylazine ratio of 1:18 was selected as a low dose for the present study, as it represents approximately half the ratio commonly used in the literature [27,28]. Following atipamezole administration, pigs showed a decreased time to recovery, as demonstrated by faster recovery score progression, faster return of palpebral and pedal withdrawal reflexes, and a higher frequency of eye blinks at an earlier timepoint. These findings align with similar research by Kim et al. (2007), who assessed the recovery of pigs following administration of XTZ at the same dose as the present study, with and without the use of a standard dose of antagonist (yohimbine) [29]. The authors showed that pigs administered antagonist had a significantly faster recovery (approximately 25 min) compared to pigs not administered antagonist, as reflected by more rapid recovery score progression (faster return to sternal recumbency, sitting and standing) [29]. Similarly, Bunnag et al. 2023 recorded the return of palpebral reflexes in pigs administered atipamezole after anaesthesia with XTZ and ketamine (1:9 ratio of atipamezole to xylazine) [27]. The authors noted a return of palpebral reflexes approximately 15 min after the administration of atipamezole [28], the same time observed in the present study when pigs were administered a 1:18 ratio of atipamezole to xylazine. This indicates that the use of low-dose atipamezole for xylazine antagonism in pigs has an effect on recovery time that is similar to the higher doses described in the literature. It is important to note that in the present study, two out of eight pigs did not have a reduced recovery time when administered atipamezole. This suggests that while a low dose of atipamezole leads to an overall reduction in recovery time in XTZ anaesthetised pigs, this reduction is subject to individual variation and may not always be a reliable outcome in all pigs. A limitation of the present study is the relatively low number of pigs assessed; therefore, it is important to note that a larger sample size may yield different results.

In the present study, six out of eight pigs administered atipamezole maintained their core temperature within normal published ranges [26]. However, when anaesthetised in the absence of atipamezole, their temperature dropped below the normal lower limit in seven out of eight pigs during recovery at the point of return to consciousness. These differences align with previous findings, where pigs receiving atipamezole antagonist for α2-adrenergic agonist anaesthesia maintained core temperature during recovery better than pigs not receiving antagonist [23,24]. Hypothermia poses anaesthetic recovery and welfare challenges by increasing the risk of thermal discomfort, impaired pharmacodynamics, and infection [28], which can, in turn, compromise the integrity of research animal models. Previous studies in hypothermic animals have demonstrated altered cytokine production, decreased tumour necrosis factor [30], and impaired platelet function and clot formation, elevating the risk of coagulopathies [28]. The present study demonstrates the benefit of low-dose atipamezole for increasing earlier pig movement during anaesthesia recovery, therefore enhancing heat production at an earlier timepoint for most pigs during anaesthesia recovery. As a relatively low number of pigs were assessed in the present study, it is important to note that a larger sample size may yield different results. Supportive warming measures, such as blankets and heat-retaining recovery surfaces, should, therefore, be implemented for anaesthetic recoveries even when antagonists are in use. This provides a practical means of enhancing the maintenance of pig core temperature during anaesthetic recovery in the laboratory environment [31].

In addition to assessing time to recovery and clinical variables, a detailed assessment of the range and timing of behaviours in relation to consciousness is also needed to comprehensively evaluate pig welfare during anaesthesia recovery. The determination of a return to consciousness of pigs in the present study was assessed within the context of the pharmacology of the anaesthesia and antagonist used, in addition to extrapolating human studies of consciousness under dissociative anaesthesia. Given that dissociative anaesthesia only induces unresponsiveness at high doses, responses such as eye blinks and leg paddling alone are not reliable indicators of consciousness [32,33,34,35]. In a study by Sarasso et al. 2015, humans anesthetised with ketamine reported awareness of their surroundings upon becoming behaviourally responsive to stimulation, highlighting the return of mental cognition as a pivotal point during recovery from dissociative anaesthesia [36]. Whilst specialised equipment such as electroencephalography (EEG) would likely allow a more precise assessment of consciousness [37], the determination of consciousness utilised in the present study provides an alternate means of assessing consciousness where EEG equipment cannot be used. This includes where the cost of such equipment is prohibitive or under high containment laboratory conditions where the use of specialised equipment is often restricted.

Behaviours exhibited at least partly before the return of consciousness and those unlikely to cause injury or distress were determined to have a low welfare impact on pigs. None of these low-impact recovery characteristics and behaviours were observed during the control (CTL) recovery, compared to 45 instances observed in four out of eight pigs during the ATZ recovery. The presence of these behaviours in pigs administered atipamezole is consistent with previous studies. Nishimura et al. noted the presence of muscle tremors in pigs administered atipamezole after medetomidine sedation [23]. Likewise, Ellis et al. reported leg paddling in XTZ-anaesthetised pigs administered antagonist [21], and excess salivation was described by Hodgkinson in pigs administered dissociative anaesthesia alone [38]. It should be noted that whilst the above-mentioned behaviours and recovery characteristics are reported in the literature, they are commonly only reported as a casual observation or the mention of their presence or absence is omitted entirely. This may be due to a genuine absence, an omission of reporting due to not being perceived as relevant to the study, or the subtlety of the behaviours causing them to be overlooked. While these recovery characteristics are arguably benign in the context of pig welfare, other behaviours observed in the present study are likely to have a more substantial negative impact on the welfare of pigs recovering from anaesthesia.

Behaviours that were solely exhibited after the return of consciousness and that were likely to cause injury or distress were determined to have a high welfare impact on pigs. During the CTL recovery, three out of eight pigs displayed seven instances of mild adverse behaviours but no instances of moderate or severe adverse behaviours. The presence of these behaviours that can cause a high negative welfare impact is not unique to the use of atipamezole and is a known consequence of anaesthesia induction and recovery in swine administered dissociative anaesthetic as part of an anaesthetic regime. Zhang et al. described these adverse effects from anaesthesia alone after miniature pigs were anaesthetised with either tiletamine–zolazepam–xylazine or ketamine–midazolam–xylazine–sufentanil. The authors observed ataxia, uncontrolled movements, and emergence delirium in 37% of pigs recovering from tiletamine–zolazepam–xylazine, although more detailed definitions of these observed behaviours were not provided [16]. Therefore, the presentation of head rearing and dropping and falls from sitting or standing in 3/8 pigs (37%) during the CTL recovery in the present study reflects what is reported in the published literature. Also established in the literature is the higher incidence of adverse behaviours of pigs administered antagonist after induction with combined α2-agonist and dissociative anaesthesia. Adverse behaviours have been noted to occur in pigs anaesthetised with these drug classes followed by administration of atipamezole, in pigs exhibiting comparable recovery times to the present study (85–100 min to full standing recovery) [25]. The antagonism of the α2-adrenergic agonist does not affect the dissociative anaesthetics, which remain in circulation and may increase the incidence of turbulent recoveries and adverse effects [25]. This was described in a study by Ellis et al., 2019, in which wild boars were anaesthetised with four different anaesthetic combinations (medetomidine–midazolam–butorphanol, butorphanol–azaperone–medetomidine, nalbuphine–medetomidine–azaperone, and tiletamine–zolazepam–xylazine) followed by administration of an α2-antagonist [21]. The authors noted that during recovery, 100% of pigs administered tiletamine–zolazepam–xylazine (4.4 mg/kg tiletamine-zolazepam, 2.5 mg/kg xylazine) followed by an antagonist (tolazine 2.2 mg/kg) exhibited paddling, struggling to rise, falling, ataxia and incoordination [21]. However, the total time from anaesthetic administration to standing recovery is not described, and therefore, this aspect cannot be compared to the present study. Despite this, these results are comparable to the 7/8 pigs (87%) in the present study, displaying some or all of the symptoms of head rearing and dropping, falls from sitting and standing, bar biting, and forceful nosing during ATZ recovery (Figure 4). What is notable in the present study is that despite the low dose of atipamezole administered compared to published literature, these behaviours with a postulated high welfare impact still occurred in 7/8 pigs. This is in comparison to the 3/8 pigs in the absence of atipamezole. In addition to the higher frequency of all recorded adverse behaviours, the present study suggests that even at a low dose, atipamezole introduces and exacerbates a variety of behaviours that have negative welfare implications for pigs under this anaesthetic regime.

## 5. Conclusions

While the use of a low dose of atipamezole in pigs anaesthetised with xylazine/tiletamine/zolazepam (XTZ) reduced recovery time and improved thermoregulation in most pigs, its use introduced or exacerbated a range of side effects and adverse behaviours. The use of atipamezole for antagonism of XTZ pig anaesthesia is, therefore, not recommended, even at the low dose used in this study. Optimising improved, easily implemented supportive warming care for large laboratory animals during anaesthesia recovery (such as blankets and recovery surfaces that retain heat) is recommended. This may improve the maintenance of core temperature and support anaesthesia recovery, whilst avoiding the use of atipamezole and the associated adverse behaviours introduced by its use.

## Figures and Tables

**Figure 1 animals-15-00258-f001:**
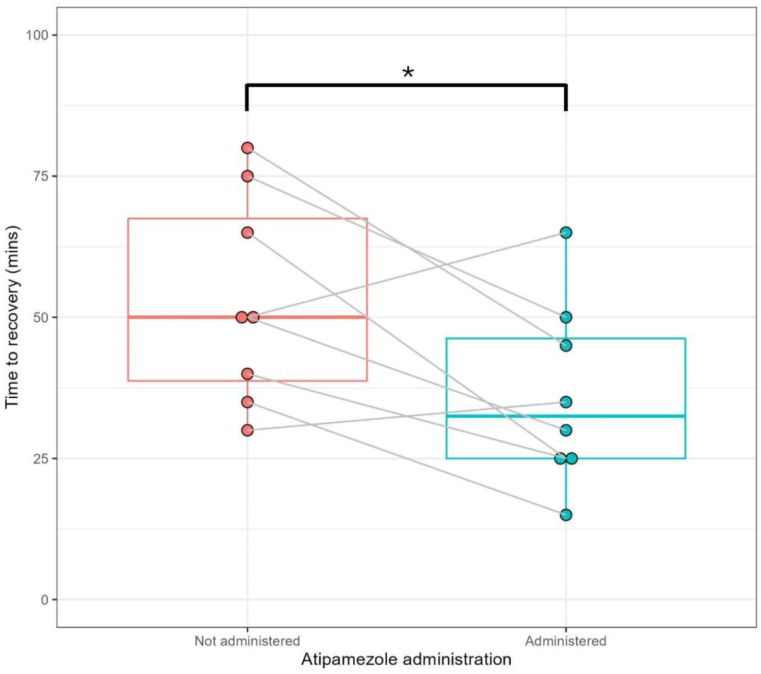
Low-dose atipamezole accelerates the return to consciousness of anaesthetised pigs. Boxplots showing the effect of low-dose atipamezole administration on the time to first reach a recovery score of 3 during anaesthesia recovery (return to consciousness). Actual time to recovery for each of the 8 individual anaesthetised pigs shown as dots and the comparative response for the individual pigs when atipamezole was administrated shown by the joining line. Centre box lines indicate the median, outer box lines represent the interquartile range. Comparison of pigs recovered without and with the use of a low dose of atipamezole (0.12 mg/kg) performed using a paired *t*-test, * *p* < 0.05.

**Figure 2 animals-15-00258-f002:**
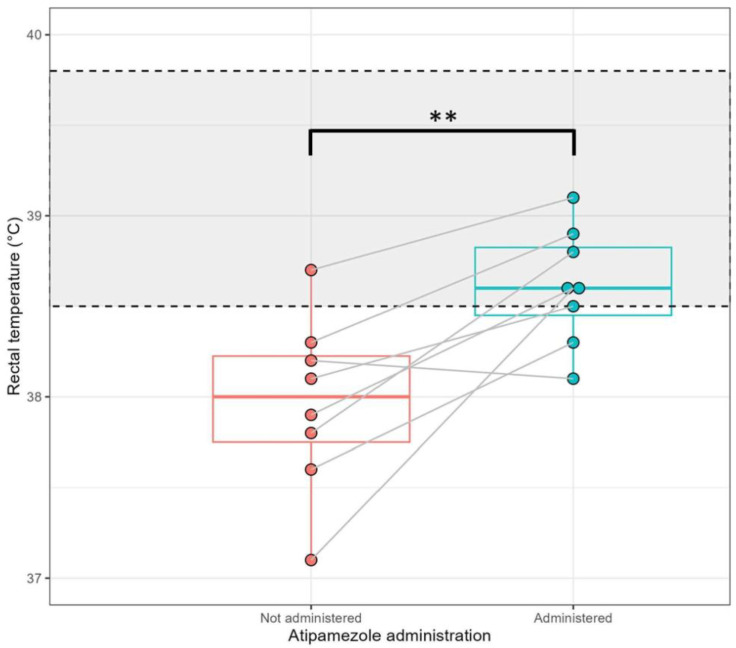
Pigs administered low-dose atipamezole show improved thermoregulation when recovering from anaesthesia. Boxplots showing the effect of low-dose atipamezole administration on rectal temperatures of anaesthetised pigs first reaching a recovery score of 3 (return to consciousness). Actual measured temperature for each of the 8 anaesthetised individual pigs shown as dots, and the comparative response for each individual pig when atipamezole was administrated shown by the joining line. Centre box lines indicate the median, outer box lines represent the interquartile range. The greyed area indicates the range of the normal rectal temperate for domestic growing pigs of comparable age and breed (38.5–39.8 °C, Sipos 2013 [26]). Anaesthesia recovery measured as the first instance of a recovery score of 3 (return to consciousness). Comparison of pigs recovered without and with the use of a low dose of atipamezole (0.12 mg/kg) performed using a paired *t*-test, ** *p* < 0.01.

**Figure 3 animals-15-00258-f003:**
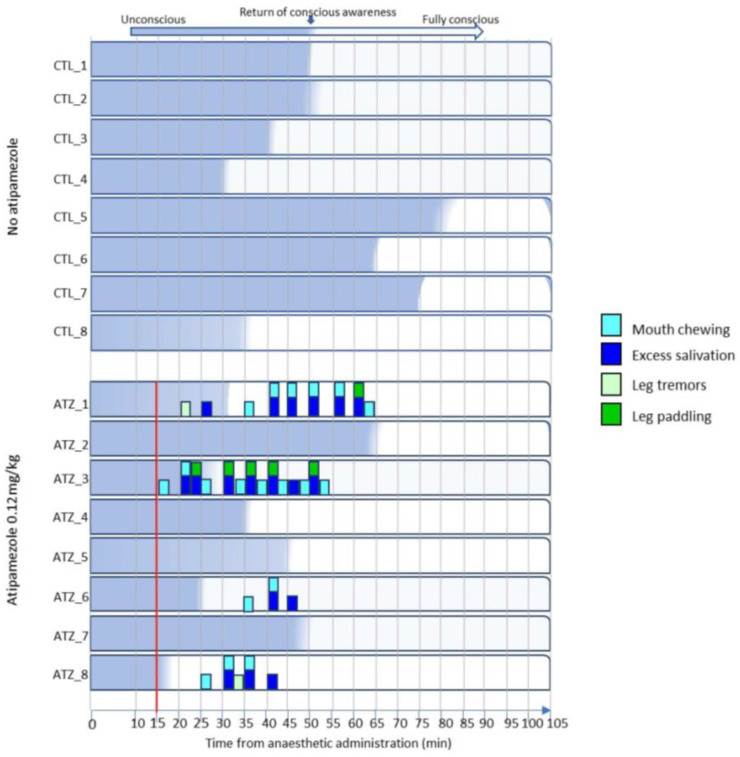
Low-dose atipamezole causes low-grade side effects and behaviours in pigs during recovery from anaesthesia. All exhibited behaviours were recorded during a 60 s period every 5 min. Bars on the graph represent the presence of the behaviour at any point during an assessment period. Conscious awareness is defined as a return of both executive function and mental cognition, equivalent to a recovery score of 3. Red vertical line represents time of atipamezole administration.

**Figure 4 animals-15-00258-f004:**
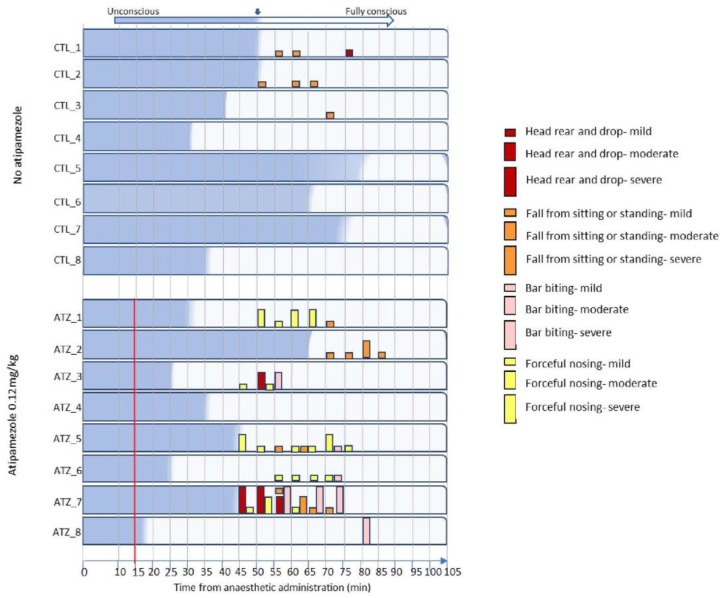
Low-dose atipamezole increases adverse behaviours in conscious pigs during recovery from anaesthesia. All exhibited behaviours were recorded during a 60 s period every 5 min. Conscious awareness is defined as a return of both executive function and mental cognition, equivalent to a recovery score of 3. Pigs received atipamezole 15 min after anaesthetic induction (atipamezole 0.12 mg/kg group). Red vertical line represents time of atipamezole administration.

**Table 1 animals-15-00258-t001:** Study timeline of eight pigs undergoing assessment of anaesthesia recovery in a JEV vaccine study.

Study Day	Study Event	Anaesthesia Administration	Atipamezole Administration	Anaesthesia Recovery Assessment
0	Arrival	No	No	N/A
0–13	Acclimation	No	No	N/A
14	Blood collected Intradermal vaccination with live attenuated JEV vaccine	Yes	No	No
63	Blood collected Intradermal challenge with JEV (10⁶ TCID_50_ ^ꝉ^)	Yes	No	No
70	Blood collected	Yes	No	Yes
77	Blood collected	Yes	Yes	Yes
85	Blood collected End of study—all pigs humanely killed ~	Yes	No	No

~ Humane killing under anaesthesia via intravenous pentobarbital up to 150 mg/kg to effect; ꝉ 50% Tissue Culture Infectious Dose.

**Table 2 animals-15-00258-t002:** Pig scoring system used for the assessment of anaesthesia recovery.

Recovery Score	Definition
0	Standing for 5 s or longer
1	Sitting on pelvic limbs and/or standing for less than 5 s
2	Keeping the position of ventral recumbency for 5 s or longer
3	Lateral recumbency with apparent spontaneous movement and response to stimulation (head lifting or purposeful limb movement)
4	Lateral recumbency with subtle, spontaneous movement (ear and nose twitching or blinking)
5	Lateral recumbency with absence of spontaneous movement (with the exception of breathing)

**Table 3 animals-15-00258-t003:** Categorisation of recovery characteristics and behaviours displayed by pigs during recovery from anaesthesia.

Recovery Characteristic or Behaviour	Definition
Eye twitch	Partial contraction of eyelid with eyes open or closed
Eye blink	Top and bottom eyelids meet then separate
Ear twitch	Ear muscle contraction and release causing visible muscle movement
Tail twitch	Any tail movement in a back/forth motion
Leg paddle	Any leg movement in a single back/forth motion
Forceful nosing	An upward nose thrust whilst nose is under the pen bar
Head rear and drop	Lifting of the head followed by hard contact with the floor
Fall from sitting or standing	Uncontrolled hard contact with the floor from a sitting or standing position
Excess salivation	Any observation of foam or pooled saliva coming from the mouth
Mouth chewing	Open or closed mouth moving in a single up/down chewing motion
Bar biting	Open mouth around a pen bar with a biting down motion

**Table 4 animals-15-00258-t004:** Categorisation of behaviours with a high welfare impact by number of times exhibited in 60 s.

Behaviour	Mild	Moderate	Severe
Head rear and drop	≤2	>2 to ≤15	>15
Fall from sitting or standing	1	≥2 to ≤5	>5
Bar biting	≤10	>10 to ≤20	>20
Forceful nosing	≤10	>10 to ≤20	>20

## Data Availability

Additional data can be found in Appendix A by visiting https://www.mdpi.com/article/10.3390/ani15020258/s1.

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
