# Peer review of "Welfare Implications of Low-Dose Atipamezole Reversal of Tiletamine/Zolazepam/Xylazine Anaesthesia in Pigs"

_animals, 2025, doi:10.3390/ani15020258_

Round 1

Reviewer 1 Report

Comments and Suggestions for Authors

Thank-you for this interesting submission, studies likely to aid welfare in animal research are indeed welcome. Overall the submission is clear and well written but contains some repetition. There are some examples of material in the incorrect section of the manuscript and perhaps some omissions in methods and discussion.

Some specific comments:

Occasional inappropriate capitalisation of the first letter of words (e.g. atipamezole line 243, pelvic in table 2)

Line 44: please check with the journal, but most journals limit to 5 key words. I might also suggest that ‘anaesthesia’ should be one ?

Line 70-76:  Although you indicate ‘procedures’ in line 78, I would suggest it is made clear in this earlier paragraph that sedation/anaesthesia  may be used for purposes other than just blood sampling in animal research, e.g. imaging studies, attending to a surgical wound/dressing etc. The specifics of your own study design of course come later

Line 100 (and ongoing): The term ‘antagonise’ may be preferable to ‘reversing’ drug effects

Lines 119-124: this part appears to move into methods, and is (in my opinion) more appropriate later. Clear definition of aims and hypothesis are the main areas to emphasise at this point of the manuscript.

Line 142: suggest ‘to aid heat retention’ rather than thermoregulation

Line 202: please could you define more clearly ‘partially present’ in relation to your palpebral reflex assessment? Does this mean a slow response?

Line 247: Suggest remove ‘the standard’

Line 248: Word ‘parametric’ can be removed. As you repeatedly compared data between sessions (each timepoint) Were data confirmed to be normally distributed, and hence appropriate to use parametric tests? Could you also clarify- line 247 indicates comparing the time to recovery of a behavioural response but the sentence goes on to describe comparing at each 5 minute time point, which would be a presence/absence of the response and hence not suitable for a t-test?

Line 257-259: It should be unnecessary to restate method at this point

Line 275: I agree with the statement of improved thermoregulation during anaesthetic recovery, but would caution against stating ‘during anaesthesia’ as this may not be true.

Line 311-313: as your definition of a high impact behaviour was one that occurred after recovery of consciousness (in methods) this statement is obvious and superfluous

Line 334: I would suggest using ‘approximately’ rather than the tilde symbol

Line 344: This part could be rephrased somewhat. The actual anaesthetic period shows similar temperatures for the animals with each treatment, however when atipamezole was not used there is ongoing heat loss after the procedure, which does not occur with atipamezole. The earlier movement etc will increase heat production. Strictly speaking this is not heat retention during anaesthesia but faster return of normal heat production in recovery. (line 356) You do not raise temperature maintenance methods (e.g. blankets) here (although mentioned in conclusion) and might be appropriate to discuss alternative ways of maintaining/raising temperature here.

Line 359-361: this appears to repeat what already has been said

Line 369: The term ‘executive function’ should be avoided in this context, as it would be impossible to perform an executive function when unconscious. There is veterinary literature commenting on the inability to assess depth of anaesthesia using palpebral reflex etc in animals that have received ketamine for anaesthesia and this would be valid to reference.

Discussion generally has a lot of repetition of methods and results

Other points it would be worth adding to discussion

(1)    sample size- you found statistically significant results, but given the variation in some animals (e.g. 2/8 did not have shorter recovery times) it is possible a larger sample could have yielded different results and although you mention the individual variation, this is not in context of sample size.

(2)    Ways to retain heat, as mentioned above

(3)    Dose justification for the atipamezole (why was decision made to use this amount)

(4)    Why papers quoted show the high impact behaviours (e.g. shorter sedative action with xylazine cf the dissociative used, hence still present at recovery)- in quoting other studies with atipamezole you do not mention the duration of anaesthesia, which will be relevant in behaviours shown in recovery

Conclusion- tramadol/multimodal analgesia not appropriate here, firstly I would not introduce a new concept to the conclusion which has not been discussed. Also the recovery would be so dependent on dose and other drugs used it is impossible to conclude without more data. Blood sampling should not be a painful procedure.

Author contributions: This is lengthy and although this is important information using initials might be both easier to read and more concise

Figures: The box on the box plots is not clearly defined- I assume the centre line is mean and the box indicating standard deviation, but is this correct?  I would recommend a higher quality image is used, as in the pdf I read the text in particular is low quality and a little unclear.

The table in figure 3 appears rather out of place and repeats figures given in text. It can be removed. Ditto figure 4.

Thank-you for this interesting submission, studies likely to aid welfare in animal research are indeed welcome. Overall the submission is clear and well written but some repetition. There are some examples of material in the incorrect section of the manuscript and perhaps some omissions in methods and discussion.

Some specific comments:

Occasional inappropriate capitalisation of the first letter of words (e.g. atipamezole line 243, pelvic in table 2)

Line 44: please check with the journal, but most journals limit to 5 key words. I might also suggest that ‘anaesthesia’ should be one ?

Line 70-76:  Although you indicate ‘procedures’ in line 78, I would suggest it is made clear in this earlier paragraph that sedation/anaesthesia  may be used for purposes other than just blood sampling in animal research, e.g. imaging studies, attending to a surgical wound/dressing etc. The specifics of your own study design of course come later

Line 100 (and ongoing): The term ‘antagonise’ may be preferable to ‘reversing’ drug effects

Lines 119-124: this part appears to move into methods, and is (in my opinion) more appropriate later. Clear definition of aims and hypothesis are the main areas to emphasise at this point of the manuscript.

Line 142: suggest ‘to aid heat retention’ rather than thermoregulation

Line 202: please could you define more clearly ‘partially present’ in relation to your palpebral reflex assessment? Does this mean a slow response?

Line 247: Suggest remove ‘the standard’

Line 248: Word ‘parametric’ can be removed. As you repeatedly compared data between sessions (each timepoint) Were data confirmed to be normally distributed, and hence appropriate to use parametric tests? Could you also clarify- line 247 indicates comparing the time to recovery of a behavioural response but the sentence goes on to describe comparing at each 5 minute time point, which would be a presence/absence of the response and hence not suitable for a t-test?

Line 257-259: It should be unnecessary to restate method at this point

Line 275: I agree with the statement of improved thermoregulation during anaesthetic recovery, but would caution against stating ‘during anaesthesia’ as this may not be true.

Line 311-313: as your definition of a high impact behaviour was one that occurred after recovery of consciousness (in methods) this statement is obvious and superfluous

Line 334: I would suggest using ‘approximately’ rather than the tilde symbol

Line 344: This part could be rephrased somewhat. The actual anaesthetic period shows similar temperatures for the animals with each treatment, however when atipamezole was not used there is ongoing heat loss after the procedure, which does not occur with atipamezole. The earlier movement etc will increase heat production. Strictly speaking this is not heat retention during anaesthesia but faster return of normal heat production in recovery. (line 356) You do not raise temperature maintenance methods (e.g. blankets) here (although mentioned in conclusion) and might be appropriate to discuss alternative ways of maintaining/raising temperature here.

Line 359-361: this appears to repeat what already has been said

Line 369: The term ‘executive function’ should be avoided in this context, as it would be impossible to perform an executive function when unconscious. There is veterinary literature commenting on the inability to assess depth of anaesthesia using palpebral reflex etc in animals that have received ketamine for anaesthesia and this would be valid to reference.

Discussion generally has a lot of repetition of methods and results

Other points it would be worth adding to discussion

(1)    sample size- you found statistically significant results, but given the variation in some animals (e.g. 2/8 did not have shorter recovery times) it is possible a larger sample could have yielded different results and although you mention the individual variation, this is not in context of sample size.

(2)    Ways to retain heat, as mentioned above

(3)    Dose justification for the atipamezole (why was decision made to use this amount)

(4)    Why papers quoted show the high impact behaviours (e.g. shorter sedative action with xylazine cf the dissociative used, hence still present at recovery)- in quoting other studies with atipamezole you do not mention the duration of anaesthesia, which will be relevant in behaviours shown in recovery

Conclusion- tramadol/multimodal analgesia not appropriate here, firstly I would not introduce a new concept to the conclusion which has not been discussed. Also the recovery would be so dependent on dose and other drugs used it is impossible to conclude without more data. Blood sampling should not be a painful procedure.

Author contributions: This is lengthy and although this is important information using initials might be both easier to read and more concise

Figures: The box on the box plots is not clearly defined- I assume the centre line is mean and the box indicating standard deviation, but is this correct?  I would recommend a higher quality image is used, as in the pdf I read the text in particular is low quality and a little unclear.

The table in figure 3 appears rather out of place and repeats figures given in text. It can be removed. Ditto figure 4.

Thank-you for this interesting submission, studies likely to aid welfare in animal research are indeed welcome. Overall the submission is clear and well written but some repetition. There are some examples of material in the incorrect section of the manuscript and perhaps some omissions in methods and discussion.

Some specific comments:

Occasional inappropriate capitalisation of the first letter of words (e.g. atipamezole line 243, pelvic in table 2)

Line 44: please check with the journal, but most journals limit to 5 key words. I might also suggest that ‘anaesthesia’ should be one ?

Line 70-76:  Although you indicate ‘procedures’ in line 78, I would suggest it is made clear in this earlier paragraph that sedation/anaesthesia  may be used for purposes other than just blood sampling in animal research, e.g. imaging studies, attending to a surgical wound/dressing etc. The specifics of your own study design of course come later

Line 100 (and ongoing): The term ‘antagonise’ may be preferable to ‘reversing’ drug effects

Lines 119-124: this part appears to move into methods, and is (in my opinion) more appropriate later. Clear definition of aims and hypothesis are the main areas to emphasise at this point of the manuscript.

Line 142: suggest ‘to aid heat retention’ rather than thermoregulation

Line 202: please could you define more clearly ‘partially present’ in relation to your palpebral reflex assessment? Does this mean a slow response?

Line 247: Suggest remove ‘the standard’

Line 248: Word ‘parametric’ can be removed. As you repeatedly compared data between sessions (each timepoint) Were data confirmed to be normally distributed, and hence appropriate to use parametric tests? Could you also clarify- line 247 indicates comparing the time to recovery of a behavioural response but the sentence goes on to describe comparing at each 5 minute time point, which would be a presence/absence of the response and hence not suitable for a t-test?

Line 257-259: It should be unnecessary to restate method at this point

Line 275: I agree with the statement of improved thermoregulation during anaesthetic recovery, but would caution against stating ‘during anaesthesia’ as this may not be true.

Line 311-313: as your definition of a high impact behaviour was one that occurred after recovery of consciousness (in methods) this statement is obvious and superfluous

Line 334: I would suggest using ‘approximately’ rather than the tilde symbol

Line 344: This part could be rephrased somewhat. The actual anaesthetic period shows similar temperatures for the animals with each treatment, however when atipamezole was not used there is ongoing heat loss after the procedure, which does not occur with atipamezole. The earlier movement etc will increase heat production. Strictly speaking this is not heat retention during anaesthesia but faster return of normal heat production in recovery. (line 356) You do not raise temperature maintenance methods (e.g. blankets) here (although mentioned in conclusion) and might be appropriate to discuss alternative ways of maintaining/raising temperature here.

Line 359-361: this appears to repeat what already has been said

Line 369: The term ‘executive function’ should be avoided in this context, as it would be impossible to perform an executive function when unconscious. There is veterinary literature commenting on the inability to assess depth of anaesthesia using palpebral reflex etc in animals that have received ketamine for anaesthesia and this would be valid to reference.

Discussion generally has a lot of repetition of methods and results

Other points it would be worth adding to discussion

(1)    sample size- you found statistically significant results, but given the variation in some animals (e.g. 2/8 did not have shorter recovery times) it is possible a larger sample could have yielded different results and although you mention the individual variation, this is not in context of sample size.

(2)    Ways to retain heat, as mentioned above

(3)    Dose justification for the atipamezole (why was decision made to use this amount)

(4)    Why papers quoted show the high impact behaviours (e.g. shorter sedative action with xylazine cf the dissociative used, hence still present at recovery)- in quoting other studies with atipamezole you do not mention the duration of anaesthesia, which will be relevant in behaviours shown in recovery

Conclusion- tramadol/multimodal analgesia not appropriate here, firstly I would not introduce a new concept to the conclusion which has not been discussed. Also the recovery would be so dependent on dose and other drugs used it is impossible to conclude without more data. Blood sampling should not be a painful procedure.

Author contributions: This is lengthy and although this is important information using initials might be both easier to read and more concise

Figures: The box on the box plots is not clearly defined- I assume the centre line is mean and the box indicating standard deviation, but is this correct?  I would recommend a higher quality image is used, as in the pdf I read the text in particular is low quality and a little unclear.

The table in figure 3 appears rather out of place and repeats figures given in text. It can be removed. Ditto figure 4.

Thank-you for this interesting submission, studies likely to aid welfare in animal research are indeed welcome. Overall the submission is clear and well written but some repetition. There are some examples of material in the incorrect section of the manuscript and perhaps some omissions in methods and discussion.

Some specific comments:

Occasional inappropriate capitalisation of the first letter of words (e.g. atipamezole line 243, pelvic in table 2)

Line 44: please check with the journal, but most journals limit to 5 key words. I might also suggest that ‘anaesthesia’ should be one ?

Line 70-76:  Although you indicate ‘procedures’ in line 78, I would suggest it is made clear in this earlier paragraph that sedation/anaesthesia  may be used for purposes other than just blood sampling in animal research, e.g. imaging studies, attending to a surgical wound/dressing etc. The specifics of your own study design of course come later

Line 100 (and ongoing): The term ‘antagonise’ may be preferable to ‘reversing’ drug effects

Lines 119-124: this part appears to move into methods, and is (in my opinion) more appropriate later. Clear definition of aims and hypothesis are the main areas to emphasise at this point of the manuscript.

Line 142: suggest ‘to aid heat retention’ rather than thermoregulation

Line 202: please could you define more clearly ‘partially present’ in relation to your palpebral reflex assessment? Does this mean a slow response?

Line 247: Suggest remove ‘the standard’

Line 248: Word ‘parametric’ can be removed. As you repeatedly compared data between sessions (each timepoint) Were data confirmed to be normally distributed, and hence appropriate to use parametric tests? Could you also clarify- line 247 indicates comparing the time to recovery of a behavioural response but the sentence goes on to describe comparing at each 5 minute time point, which would be a presence/absence of the response and hence not suitable for a t-test?

Line 257-259: It should be unnecessary to restate method at this point

Line 275: I agree with the statement of improved thermoregulation during anaesthetic recovery, but would caution against stating ‘during anaesthesia’ as this may not be true.

Line 311-313: as your definition of a high impact behaviour was one that occurred after recovery of consciousness (in methods) this statement is obvious and superfluous

Line 334: I would suggest using ‘approximately’ rather than the tilde symbol

Line 344: This part could be rephrased somewhat. The actual anaesthetic period shows similar temperatures for the animals with each treatment, however when atipamezole was not used there is ongoing heat loss after the procedure, which does not occur with atipamezole. The earlier movement etc will increase heat production. Strictly speaking this is not heat retention during anaesthesia but faster return of normal heat production in recovery. (line 356) You do not raise temperature maintenance methods (e.g. blankets) here (although mentioned in conclusion) and might be appropriate to discuss alternative ways of maintaining/raising temperature here.

Line 359-361: this appears to repeat what already has been said

Line 369: The term ‘executive function’ should be avoided in this context, as it would be impossible to perform an executive function when unconscious. There is veterinary literature commenting on the inability to assess depth of anaesthesia using palpebral reflex etc in animals that have received ketamine for anaesthesia and this would be valid to reference.

Discussion generally has a lot of repetition of methods and results

Other points it would be worth adding to discussion

(1)    sample size- you found statistically significant results, but given the variation in some animals (e.g. 2/8 did not have shorter recovery times) it is possible a larger sample could have yielded different results and although you mention the individual variation, this is not in context of sample size.

(2)    Ways to retain heat, as mentioned above

(3)    Dose justification for the atipamezole (why was decision made to use this amount)

(4)    Why papers quoted show the high impact behaviours (e.g. shorter sedative action with xylazine cf the dissociative used, hence still present at recovery)- in quoting other studies with atipamezole you do not mention the duration of anaesthesia, which will be relevant in behaviours shown in recovery

Conclusion- tramadol/multimodal analgesia not appropriate here, firstly I would not introduce a new concept to the conclusion which has not been discussed. Also the recovery would be so dependent on dose and other drugs used it is impossible to conclude without more data. Blood sampling should not be a painful procedure.

Author contributions: This is lengthy and although this is important information using initials might be both easier to read and more concise

Figures: The box on the box plots is not clearly defined- I assume the centre line is mean and the box indicating standard deviation, but is this correct?  I would recommend a higher quality image is used, as in the pdf I read the text in particular is low quality and a little unclear.

The table in figure 3 appears rather out of place and repeats figures given in text. It can be removed. Ditto figure 4.

Thank-you for this interesting submission, studies likely to aid welfare in animal research are indeed welcome. Overall the submission is clear and well written but some repetition. There are some examples of material in the incorrect section of the manuscript and perhaps some omissions in methods and discussion.

Some specific comments:

Occasional inappropriate capitalisation of the first letter of words (e.g. atipamezole line 243, pelvic in table 2)

Line 44: please check with the journal, but most journals limit to 5 key words. I might also suggest that ‘anaesthesia’ should be one ?

Line 70-76:  Although you indicate ‘procedures’ in line 78, I would suggest it is made clear in this earlier paragraph that sedation/anaesthesia  may be used for purposes other than just blood sampling in animal research, e.g. imaging studies, attending to a surgical wound/dressing etc. The specifics of your own study design of course come later

Line 100 (and ongoing): The term ‘antagonise’ may be preferable to ‘reversing’ drug effects

Lines 119-124: this part appears to move into methods, and is (in my opinion) more appropriate later. Clear definition of aims and hypothesis are the main areas to emphasise at this point of the manuscript.

Line 142: suggest ‘to aid heat retention’ rather than thermoregulation

Line 202: please could you define more clearly ‘partially present’ in relation to your palpebral reflex assessment? Does this mean a slow response?

Line 247: Suggest remove ‘the standard’

Line 248: Word ‘parametric’ can be removed. As you repeatedly compared data between sessions (each timepoint) Were data confirmed to be normally distributed, and hence appropriate to use parametric tests? Could you also clarify- line 247 indicates comparing the time to recovery of a behavioural response but the sentence goes on to describe comparing at each 5 minute time point, which would be a presence/absence of the response and hence not suitable for a t-test?

Line 257-259: It should be unnecessary to restate method at this point

Line 275: I agree with the statement of improved thermoregulation during anaesthetic recovery, but would caution against stating ‘during anaesthesia’ as this may not be true.

Line 311-313: as your definition of a high impact behaviour was one that occurred after recovery of consciousness (in methods) this statement is obvious and superfluous

Line 334: I would suggest using ‘approximately’ rather than the tilde symbol

Line 344: This part could be rephrased somewhat. The actual anaesthetic period shows similar temperatures for the animals with each treatment, however when atipamezole was not used there is ongoing heat loss after the procedure, which does not occur with atipamezole. The earlier movement etc will increase heat production. Strictly speaking this is not heat retention during anaesthesia but faster return of normal heat production in recovery. (line 356) You do not raise temperature maintenance methods (e.g. blankets) here (although mentioned in conclusion) and might be appropriate to discuss alternative ways of maintaining/raising temperature here.

Line 359-361: this appears to repeat what already has been said

Line 369: The term ‘executive function’ should be avoided in this context, as it would be impossible to perform an executive function when unconscious. There is veterinary literature commenting on the inability to assess depth of anaesthesia using palpebral reflex etc in animals that have received ketamine for anaesthesia and this would be valid to reference.

Discussion generally has a lot of repetition of methods and results

Other points it would be worth adding to discussion

(1)    sample size- you found statistically significant results, but given the variation in some animals (e.g. 2/8 did not have shorter recovery times) it is possible a larger sample could have yielded different results and although you mention the individual variation, this is not in context of sample size.

(2)    Ways to retain heat, as mentioned above

(3)    Dose justification for the atipamezole (why was decision made to use this amount)

(4)    Why papers quoted show the high impact behaviours (e.g. shorter sedative action with xylazine cf the dissociative used, hence still present at recovery)- in quoting other studies with atipamezole you do not mention the duration of anaesthesia, which will be relevant in behaviours shown in recovery

Conclusion- tramadol/multimodal analgesia not appropriate here, firstly I would not introduce a new concept to the conclusion which has not been discussed. Also the recovery would be so dependent on dose and other drugs used it is impossible to conclude without more data. Blood sampling should not be a painful procedure.

Author contributions: This is lengthy and although this is important information using initials might be both easier to read and more concise

Figures: The box on the box plots is not clearly defined- I assume the centre line is mean and the box indicating standard deviation, but is this correct?  I would recommend a higher quality image is used, as in the pdf I read the text in particular is low quality and a little unclear.

The table in figure 3 appears rather out of place and repeats figures given in text. It can be removed. Ditto figure 4.

Author Response

Comment 1: Thank-you for this interesting submission, studies likely to aid welfare in animal research are indeed welcome. Overall the submission is clear and well written but contains some repetition. There are some examples of material in the incorrect section of the manuscript and perhaps some omissions in methods and discussion.

Response 1: My sincerest thanks to Reviewer 1 for their kind words, useful feedback, and quick review of this manuscript- particularly given the busy time of year.  The feedback you have provided is invaluable to me as a young researcher in the field, I really appreciate it.

Some specific comments:

Comment 2: Occasional inappropriate capitalisation of the first letter of words (e.g. atipamezole line 243, pelvic in table 2)

Response 2: Thank you, all inappropriate capitalisations have been corrected

Comment 3: Line 44: please check with the journal, but most journals limit to 5 key words. I might also suggest that ‘anaesthesia’ should be one ?

Response 3: Thank you for this suggestion, ‘anaesthesia’ has been added as a keyword and ‘recovery’ and ‘adverse’ have been removed (Line 44)

Comment 4: Line 70-76:  Although you indicate ‘procedures’ in line 78, I would suggest it is made clear in this earlier paragraph that sedation/anaesthesia may be used for purposes other than just blood sampling in animal research, e.g. imaging studies, attending to a surgical wound/dressing etc. The specifics of your own study design of course come later

Response 4: Thank you, the following sentence has been added to specify the wider range of reasons for the use of anaesthesia: ‘Anaesthesia can be useful for a range of procedures in pigs, including blood collection, restraint for imaging studies, and wound management’ (Line 76), with references added to support this statement (references [1,10] (Line 77)

Comment 5: Line 100 (and ongoing): The term ‘antagonise’ may be preferable to ‘reversing’ drug effects

Response 5: Thank you for this suggestion. ‘Reversal’ and ‘reverse’ has been replaced with ‘antagonism’ and ‘antagonise’ throughout the manuscript. The exception is the simple summary (Line 13), where the use of ‘reversal’ and ‘reverse’ is maintained for a lay audience.

Comment 6: Lines 119-124: this part appears to move into methods, and is (in my opinion) more appropriate later. Clear definition of aims and hypothesis are the main areas to emphasise at this point of the manuscript.

Response 6: Thank you for this suggestion, this has been removed (Lines 118-124)

Comment 7: Line 142: suggest ‘to aid heat retention’ rather than thermoregulation

Response 7: Thank you, ‘thermoregulation’ has been replaced with ‘heat retention’ as suggested (Line 145)

Comment 8: Line 202: please could you define more clearly ‘partially present’ in relation to your palpebral reflex assessment? Does this mean a slow response?

Response 8: Thank you, we agree this further definition is valuable. The definition has been updated to partially present ‘as a slower response and incomplete closure of the eyelid’ (Line 207)

Comment 9: Line 247: Suggest remove ‘the standard’

Response 9: Thank you, ‘the standard’ has been removed (Line 254)

Comment 10: Line 248: Word ‘parametric’ can be removed. As you repeatedly compared data between sessions (each timepoint) Were data confirmed to be normally distributed, and hence appropriate to use parametric tests? Could you also clarify- line 247 indicates comparing the time to recovery of a behavioural response but the sentence goes on to describe comparing at each 5 minute time point, which would be a presence/absence of the response and hence not suitable for a t-test?

Response 10: Thank you, ‘parametric’ has been removed (Line 256). Data for both time to recovery and temperature were found to be normally distributed, passing normality tests via both the D’Agostino and Pearson test and the Shapiro-Wilk test (alpha=0.05). This has been added to the manuscript (Line 247). Clarification has been added regarding the time to recovery assessment at Line 255, which hopefully clarifies that statistical analysis was conducted on the time in minutes for each individual pig to reach a recovery score of 3 (rather than a presence/ absence), therefore making a t-test suitable for this analysis.

Comment 11: Line 257-259: It should be unnecessary to restate method at this point

Response 11: Thank you, this has been amended to: ‘To determine the impact of low-dose atipamezole on the recovery of pigs anaesthetised with XTZ, the recovery of pigs from anaesthesia with and without a low dose of atipamezole was assessed’ (Line 264). We feel that this provides a brief summary of study context, whilst addressing the concern of restating methods.

Comment 12: Line 275: I agree with the statement of improved thermoregulation during anaesthetic recovery, but would caution against stating ‘during anaesthesia’ as this may not be true.

Response 12: Thank you, we agree that ‘during anaesthetic recovery’ is more appropriate- this has been amended at Line 292

Comment 13: Line 311-313: as your definition of a high impact behaviour was one that occurred after recovery of consciousness (in methods) this statement is obvious and superfluous

Response 13: Thank you, we agree with this statement and have removed this at Line 353.

Comment 14: Line 334: I would suggest using ‘approximately’ rather than the tilde symbol

Response 14: Thank you for this suggestion, the tilde symbol has been replaced with ‘approximately at Lines 378 and 384.

Comment 15: Line 344: This part could be rephrased somewhat. The actual anaesthetic period shows similar temperatures for the animals with each treatment, however when atipamezole was not used there is ongoing heat loss after the procedure, which does not occur with atipamezole. The earlier movement etc will increase heat production. Strictly speaking this is not heat retention during anaesthesia but faster return of normal heat production in recovery. (line 356) You do not raise temperature maintenance methods (e.g. blankets) here (although mentioned in conclusion) and might be appropriate to discuss alternative ways of maintaining/raising temperature here.

Response 15: Thank you for this suggestion, we agree that this is more accurate. This paragraph has been re-written to state ‘anaesthetic recovery’ (Lines 397, 399, 406, 407, and 408). In addition, the following has been added at line 415 with a reference supporting the statement: ‘The use of supportive warming measures, such as blankets, should therefore be implemented for anaesthetic recoveries even when antagonists are in use. This provides a practical means of enhancing the maintenance of pig core temperature during anaesthetic recovery in the laboratory environment [32].’ Reference added at Line 573: Dent, B. T., Stevens, K. A., & Clymer, J. W. (2016). Forced-air warming provides better control of body temperature in porcine surgical patients. Veterinary Sciences3(3),22. https://doi.org/10.3390/vetsci3030022

The numbering of subsequent references have also been updated to reflect the introduction of this new reference.

Comment 16: Line 359-361: this appears to repeat what already has been said

Response 16: Thank you, we agree and this paragraph has been revised as described in Response 15.

Comment 17: Line 369: The term ‘executive function’ should be avoided in this context, as it would be impossible to perform an executive function when unconscious. There is veterinary literature commenting on the inability to assess depth of anaesthesia using palpebral reflex etc in animals that have received ketamine for anaesthesia and this would be valid to reference.

Response 17: Thank you, this is valuable clarification and literature. Reference to executive function has been removed (Line 425) and veterinary references regarding assessment of anaesthetic depth under dissociative anaesthesia (ketamine and tiletamine) using responses such as palpebral and pedal reflexes have been reviewed and added (Line 427). These have also been added to the reference section, and subsequent reference numbering updated. References as follows:

-Haskins, S. C. (1992). General guidelines for judging anesthetic depth. The Veterinary clinics of North America. Small animal practice22(2), 432-434. https://doi.org/10.1016/s0195-5616(92)50659-3

-Lin, H. C., Thurmon, J. C., Benson, G. J., & Tranquilli, W. J. (1993). Telazol-a review of its pharmacology and use in veterinary medicine. https://doi.org/10.1111/j.1365-2885.1993.tb00206.x

-Gross, M. E., & Pablo, L. S. (2015). Ophthalmic patients. Veterinary Anesthesia and Analgesia: The Fifth Edition of Lumb and Jones, 961-982. https://doi.org/10.1002/9781119421375.ch52

Comment 18: Discussion generally has a lot of repetition of methods and results

Response 18: Thank you for this observation, we agree and have removed the following repetition in the discussion: Line 410 ‘However, it is important to note that one out of eight pigs in the present study did not maintain rectal temperatures within the normal range even with the administration of atipamezole. This suggests that while a low dose of atipamezole leads to an overall improvement in thermoregulation in XTZ anaesthetised pigs, this is subject to individual variation and may not always be a reliable out-come in all pigs.’

Line 430 ‘Following this premise, we defined a recovery score of 3 in the present study as the stage of recovery where pigs displayed signs of both executive functioning (such as movement and disconnected interaction with the environment) and mental cognition (ability to understand, process and respond to stimuli). This was postulated as the most likely point of return to consciousness’.

Line 461 ‘These behaviours were head rearing and dropping, falling from sitting or standing, bar biting and forceful nosing. High-impact adverse behaviours were classified as mild, moderate or severe displays based on the number of instances displayed at each assessment period’.

Other points it would be worth adding to discussion

Comment 19: (1)    sample size- you found statistically significant results, but given the variation in some animals (e.g. 2/8 did not have shorter recovery times) it is possible a larger sample could have yielded different results and although you mention the individual variation, this is not in context of sample size.

Response 19: Thank you, the following has been added to line 392 ‘A limitation of the present study is the relatively low number pigs assessed; therefore it is important to note that a larger sample size may yield different results’ and line 408 ‘As a relatively low number of pigs were assessed in the present study, it is important to note that a larger sample size may yield different results’.

Comment 20: Ways to retain heat, as mentioned above

Response 20: Thank you, the following addition has been made at line 414 ‘The use of supportive warming measures, such as blankets and heat-retaining recovery surfaces, should therefore be implemented for anaesthetic recoveries even when antagonists are in use. This provides a practical means of enhancing the maintenance of pig core temperature during anaesthetic recovery in the laboratory environment [32].’ Reference added at Line 631: Dent, B. T., Stevens, K. A., & Clymer, J. W. (2016). Forced-air warming provides better control of body temperature in porcine surgical patients. Veterinary Sciences3(3),22. https://doi.org/10.3390/vetsci3030022

Comment 21: Dose justification for the atipamezole (why was decision made to use this amount)

Response 21: Thank you, the following has been added to line 370 ‘An atipamezole to xylazine ratio of 1:18 was selected as a low dose for the present study, as it represents approximately half the ratio commonly used in literature [28,29]’.

Comment 22: Why papers quoted show the high impact behaviours (e.g. shorter sedative action with xylazine cf the dissociative used, hence still present at recovery)- in quoting other studies with atipamezole you do not mention the duration of anaesthesia, which will be relevant in behaviours shown in recovery

Response 22: Thank you for this valuable suggestion. The total time to recovery from anaesthetic injection or succumbing to anaesthesia to standing recovery is not detailed in the study by Ellis et al., 2019 (only the time to succumb to anaesthesia, followed by the time to recovery after the administration of antagonist). The time between these events is not described. In light of this, the following sentence in regard to this study has been added at line 490 ‘However, the total time from anaesthetic administration to standing recovery is not described, and therefore this aspect cannot be compared to the present study’. Additionally, the following has been added at line 478 in relation to the study by Morelli et al., 2021 ‘These adverse behaviours have been noted to occur in pigs anaesthetised with these drug classes followed by administration of atipamezole, in pigs exhibiting comparable recovery times to the present study (85-100 minutes to full standing recovery) [25].

Comment 23: Conclusion- tramadol/multimodal analgesia not appropriate here, firstly I would not introduce a new concept to the conclusion which has not been discussed. Also the recovery would be so dependent on dose and other drugs used it is impossible to conclude without more data. Blood sampling should not be a painful procedure.

Response 23: Thank you for your advice here, the following has been removed from line 508 ‘To mitigate these recovery characteristics and behaviours introduced or exacerbated by atipamezole, the incorporation of more multi-modal anaesthesia, such as the addition of tramadol or butorphanol, could be explored. However, caution should be exercised to ensure that these potential benefits are not outweighed by the introduction of other variables, such as increased time to recovery and the introduction of further physiological effects’.

Comment 24: Author contributions: This is lengthy and although this is important information using initials might be both easier to read and more concise.

Response 24: Thank you for this. This format is specified by the journal, therefore we have left the author contributions as written. However, we are happy to revisit this if preferable/acceptable to the journal editor.

Comment 25: Figures: The box on the box plots is not clearly defined- I assume the centre line is mean and the box indicating standard deviation, but is this correct?  I would recommend a higher quality image is used, as in the pdf I read the text in particular is low quality and a little unclear.

Response 25: Thank you for bringing this omission in the figure legends to our attention. Specification that the centre line is the median and the outer box the interquartile range has been added to figure legends 1 (line 286) and 2 (line 317). We have also re-added all figures in JPEG format for higher resolution and readability.

Comment 26: The table in figure 3 appears rather out of place and repeats figures given in text. It can be removed. Ditto figure 4.

Response 26: Thank you, we agree with your suggestion here and have removed the tables in both figure 3 and figure 4.

Once again, our sincerest gratitude to Reviewer 1 for lending their time and expertise to improving this manuscript.

Reviewer 2 Report

Comments and Suggestions for Authors

Dear Authors

The manuscript is clear and very interesing. The experimental desing is appropriate. The conclusions are presented well and align with the results obtained during the investigation.

I have got a few comment to the text:

L43: Please reconsider keywords: recovery and adverse.

L50: Can you specify other biological samples?

L55: Please remove the line. The comment applies to the entire manuscript: LL 77, 95, 111, etc.

L56: Please be consistent with punctuation: use eirher e.g., or just e.g. (no comma, like in L191, L192, L193).

L64: Isn't it [7-9] or [7,8,9]? Please verify the requirements.

L107: e.g. not eg

L112: Please specify the vaccination protocol.

L114: Please remove the spacing error after the stroke (tiletamine/zolazepam).

L129: Please define the body weight.

L131: Please specify the health check.

L144: Please add the information on the viral shedding. What did you do exactly to analyse it?

L155: Isn't it Table 1. instead of Table 1: ? Please verify it.

L155: Please correct the spacing error (End of study-).

L155, 156: Please reconsider using the term humane killing. Why not euthanasia?

L160: Please add the information on the manufacturer.

L161: When did the first assessment take place? The pigs were 6 weeks old at the beggining of the trial.

L181: pelvic, not Pelvic

L218: Please remove the spacing errors (after the strokes).

L243: atipamezole, not Atipamezole

L272: There are two asterisks. Please compare it with the Figure 1 (only one). Also, please correct the justification. It applies to all the other figures.

L301: Please remove the spacing error (anaesthesia .)

L309: Please remove the spacing error.

L390: Is the year acceptable? Please verify.

L443: Can your conclusions be extended to the usage of these substances in commercially reared swine?

Best regards

Author Response

Comment 1: The manuscript is clear and very interesing. The experimental desing is appropriate. The conclusions are presented well and align with the results obtained during the investigation.

Response 1: My sincerest thanks to Reviewer 2 for their kind words, useful feedback, and quick review of this manuscript- particularly given the busy time of year.  The feedback you have provided is invaluable to me as a young researcher in the field, I really appreciate it.

Comment 2: L43: Please reconsider keywords: recovery and adverse.

Response 2: Thank you, recovery and adverse have been removed and 'anaesthesia' added as a keyword (Line 44)

Comment 3: L50: Can you specify other biological samples? 

Response 3: Thank you, we agree that would be useful for clarity. This has been amended to '...and other biological samples (such as swabs) ...' (Line 50)

Comment 4: L55: Please remove the line. The comment applies to the entire manuscript: LL 77, 95, 111, etc.

Response 4: Thank you, all lines between paragraphs throughout the Introduction and Discussion have been removed

Comment 5: L56: Please be consistent with punctuation: use eirher e.g., or just e.g. (no comma, like in L191, L192, L193).

Response 5: Thank you for detecting this inconsistency, all 'e.g.' have been changed to 'e.g.,' to ensure consistency (Lines 104, 191, 192 and 193)

Comment 6: L64: Isn't it [7-9] or [7,8,9]? Please verify the requirements.

Response 6: Thank you for detecting this. Multiple references have now been changed to (for example) [7,8,9] as per journal requirements (Line 62, 80, 81, 91, 96, 102 and 400)

Comment 7: L107: e.g. not eg

Response 7: Thank you, eg has been changed to e.g., for consistency throughout manuscript and at Line 104

Comment 8: L112: Please specify the vaccination protocol.

Response 8: Thank you, details of the vaccination protocol are listed in Table 1 (Line 154), which hopefully addresses this comment

Comment 9: L114: Please remove the spacing error after the stroke (tiletamine/zolazepam).

Response 9: Thank you for detecting this, the space has been removed to now read tiletamine/zolazepam (Line 110)

Comment 10: L129: Please define the body weight

Response 10: Thank you for this suggestion, the weights of pigs at arrival has now been added as follows: 'All pigs were weighed (21-25kg) and health checked on arrival prior to allocation to the JEV vaccine study (Line 127)

Comment 11: L131: Please specify the health check.

Response 11: Thank you, I agree with this recommendation and an additional sentence has been added as follows: The health check on arrival consisted of measurement of respiratory rate and heart rate, assessment of skin, ocular and oral health, presence or absence of scouring or abnormal faecal output, and body condition score (Line 128)

Comment 12: L144: Please add the information on the viral shedding. What did you do exactly to analyse it?

Response 12: Thank you, we agree this is a valuable addition for the context of the study. The following has been added: '...and did not shed virus via oral swab, nasal swab or blood collected daily for eight days after viral administration' (Line 143)

Comment 13: L155: Isn't it Table 1. instead of Table 1: ? Please verify it.

Response 13: Thank you, this has been corrected to 'Table 1.' (Line 154)

Comment 14: L155: Please correct the spacing error (End of study-).

Response 14: Thank you, the space after 'End of study' has been removed at Line 154

Comment 15: L155, 156: Please reconsider using the term humane killing. Why not euthanasia?

Response 15: Thank you for raising this for discussion. We have used the term humane killing as the pigs were healthy, not distressed and not suffering from any clinical signs of disease at the end of the study. Therefore, we feel that using the term euthanasia in this context would not be as reflective of the health and welfare status of pigs at the time of their death.

Comment 16: L160: Please add the information on the manufacturer.

Response 16: Thank you, this has now been added as (Virbac Animal Health, Carros, France) (Line 159)

Comment 17: L161: When did the first assessment take place? The pigs were 6 weeks old at the beggining of the trial.

Response 17: Thank you, we agree that specifying the age of pigs at the time of the assessments provides valuable context to the results. This has been amended to 'Pigs were aged 16 weeks and weighed in the range of 48-55kg at the first assessment period, and aged 17 weeks and weighed 50-57kg at the second assessment period' (Line 161)

Comment 18: L181: pelvic, not Pelvic

Response 18: Thank you for detecting this, 'Pelvic' has been corrected to pelvic (Line 187)

Comment 19: L218: Please remove the spacing errors (after the strokes).

Response 19: Thank you, line spacing of table category has been amended to single spacing (Line 218)

Comment 20: L243: atipamezole, not Atipamezole

Response 20: Thank you, this has been corrected to atipamezole (Line 245)

Comment 21: L272: There are two asterisks. Please compare it with the Figure 1 (only one). Also, please correct the justification. It applies to all the other figures.

Response 21: Thank you, the asterisk has been corrected to one asterisk and the figure legend justified (Line 307). Justification has also been done for the other figures (Line 282, 338 and 360)

Comment 22: L301: Please remove the spacing error (anaesthesia .)

Response 22: Thank you, the space has been removed in the figure legend title (Figure 3, Line 339)

Comment 23: L309: Please remove the spacing error.

Response 23: Thank you, the space has been removed (Line 351)

Comment 24: L390: Is the year acceptable? Please verify.

Response 24: Thank you for raising this for discussion. Whilst we acknowledge that this publication is from 1992, we feel that the study design, age and breed of pigs, and anaesthesia regime still make this literature relevant in discussing the results of the present study in relation to mild adverse effects observed after atipamezole administration. The low number of published papers that report on relatively minor adverse behaviours and responses to atipamezole (possibly due to omission of reporting these responses or subtle responses going unnoticed) leads us to be comfortable in including this paper, particularly given that it is not heavily referenced or relied upon in drawing our conclusions. Thank you for raising this, as it prompted us to re-check the journal guidelines and we can confirm that the age of the paper is acceptable for the journal guidelines.

Comment 25: L443: Can your conclusions be extended to the usage of these substances in commercially reared swine?

Response 25: Thank you for raising this, we agree that this is an important consideration. Due to withholding periods of using anaesthetic and antagonists for pigs reared for meat, the use of these agents is uncommon in a commercial setting. Because of this, we would prefer to maintain the focus of our conclusions to the laboratory setting, where repeat anaesthetic events are common for pigs.

Once again, our sincerest gratitude to Reviewer 2 for lending their time and expertise to improving this manuscript.

Round 2

Reviewer 1 Report

Comments and Suggestions for Authors

Thank-you for your changes which have much improved the manuscript. Please check carefully for spaces (e.g. between 'approximately' and 15) and detailed changes such as 'in the literature' [line 360].